

# Evaluating moss diversity and biomass for sustainable harvesting methods in semi-arid forests of Turkey

Serhat Ursavaş[1] and Recep Söyler[2]

[1] Department of Forest Engineering, Faculty of Forestry, Çankırı Karatekin University, Çankırı, Turkey
[2] Department of Forest Engineering, Graduate School of Natural and Applied Sciences, Çankırı Karatekin University, Çankırı, Turkey

## ABSTRACT

In Turkey, it is legal to harvest moss from designated areas; however, the lack of comprehensive inventory studies in these harvested zones poses a significant threat to moss species. Harvesting without proper inventories can negatively impact rare, sensitive, and even endemic species in the region. Furthermore, research on the sustainable amount of moss harvestable per hectare in forested areas is severely lacking. The goal of this study, which covered 4,200 hectares on Eldivan Mountain, was to close the significant gap in moss inventory and sustainable harvesting methods. Sampling was conducted every 300-meters, measuring mosses in four $m^2$ ground plots and 50 $m^2$ tree plots. The total area covered by the identified moss species was approximately 97,216,557 $m^2$, with a total dry weight of 44,640,972 kilograms. The most widespread ground species, *Syntrichia ruralis* (Hedw.) F. Weber & D. Mohr, covered 64,772,801 $m^2$ with a dry weight of 623,268 kilograms, while the dominant tree species, *Hypnum cupressiforme* var. *lacunosum* Brid., covered 3,937,266 $m^2$ with a dry weight of 1,448,533 kilograms. The research determines that the collection of epiphytic mosses is unsustainable, owing to insufficient rainfall in Turkey's semi-arid areas. We recommend a sustainable harvest rate of 1–1.5 tons per hectare for ground mosses to balance ecological conservation with commercial objectives. These findings furnish critical information for conservation strategies and the formulation of sustainable moss harvesting methodologies.

# INTRODUCTION

The global demand for non-timber forest products (NTFPs), including mosses, has surged, emphasizing the need for sustainable harvesting practices (*Peck & Christy, 2006*). While the USA leads in the commercial collection of epiphytic mosses for floriculture and horticulture, data on ground moss harvesting remains limited (*Vance & Thomas, 1997*; *Alexander, 2002*; *Jones, Mclain & Weigand, 2002*). The moss trade contributes significantly to local economies, with annual exports from regions like the Appalachian Mountains valued at $2.3 million (*Muir, 2004*; *Peck & Moldenke, 2010*). In contrast, Turkey lacks comprehensive data on moss biomass, distribution, and sustainable harvesting methods,

Corresponding author
Serhat Ursavaş,
serhatursavas@gmail.com

putting endemic and sensitive species at risk. This deficiency underscores the urgent need for inventory studies and regulatory measures to safeguard biodiversity and ensure sustainable resource management (*Ursavaş & Söyler, 2015*; *Ursavaş & Ediş, 2024*).

Annually, the USA harvests approximately 3.7 million kilograms of mosses and liverworts, with Oregon reporting biomass densities of 693 kg/ha (*Peck, 2006*; *Peck & McCune, 1998*). Turkey, by comparison, produces 184,000 kg of dried moss each year, but lacks inventory assessments to evaluate bryophyte distribution and biomass. Moss harvesting in Turkey is economically significant, particularly in the Western Black Sea and Aegean regions, yet unregulated practices jeopardize biodiversity and ecosystem health. Addressing these deficiencies is critical for the sustainable management of moss resources and the conservation of endemic species (*Ursavaş & Söyler, 2015*).

Bryophyte research in Turkey began in the 1800s, with five–ten new records documented annually by Turkish researchers. However, studies primarily focus on coastal regions, leaving the semi-arid interior and eastern areas, such as Cankiri-Eldivan Mountain, underexplored (*Ursavaş & Keçeli, 2019*). These areas may harbor rare and sensitive species, yet insufficient research and unregulated harvesting threaten biodiversity. Moss ecosystems in Turkey are vital for maintaining soil stability, water retention, and nutrient cycling, especially in forested and semi-arid regions, thus playing a crucial role in ecological balance (*Ursavaş & Söyler, 2015*; *Ursavaş et al., 2021*).

The absence of inventory studies on mosses in Turkey hinders conservation efforts and increases the risk of biodiversity loss (*Demir, 2013*). Unregulated harvesting reduces moss resources, threatening economic sectors like floriculture and landscaping while impairing ecological services. Addressing these gaps is necessary to harmonize commercial interests with ecological preservation (*Ursavaş, Birben & Albayrak, 2013*). The dataset from this study provides valuable insights for developing conservation strategies, sustainable harvesting methods, and ecosystem management plans in semi-arid and forested regions.

Sustainable moss harvesting involves extracting mosses at rates that allow natural regeneration, aligning with ecological principles of resilience and carrying capacity (*Holling, 1973*; *Odum, 1985*). Key indicators include biomass recovery rates, species composition, and habitat integrity. This study recommends regulating sustainable harvesting through biomass thresholds and rotational systems to promote recovery. The primary objectives are (1) to identify rare or sensitive bryophyte species to minimize harvesting impacts and conserve biodiversity and (2) to quantify bryophyte biomass to assess sustainable yields, ensuring long-term ecosystem health. In the context of this study, biomass refers to the total dry weight of moss species within a specified area, measured in kilograms (kg). We measure the surface areas occupied by moss species on the soil in square meters ($m^2$).

## MATERIAL AND METHODS

### Study area

Eldivan Mountain, located in Cankiri Province, Turkey (Fig. 1), encompasses 4,200 hectares and underwent slow afforestation following a big flood in 1952. Before the inundation, indigenous *Pinus nigra* subsp. *pallasiana* (Lamb. Holmboe) inhabited the area sparingly,

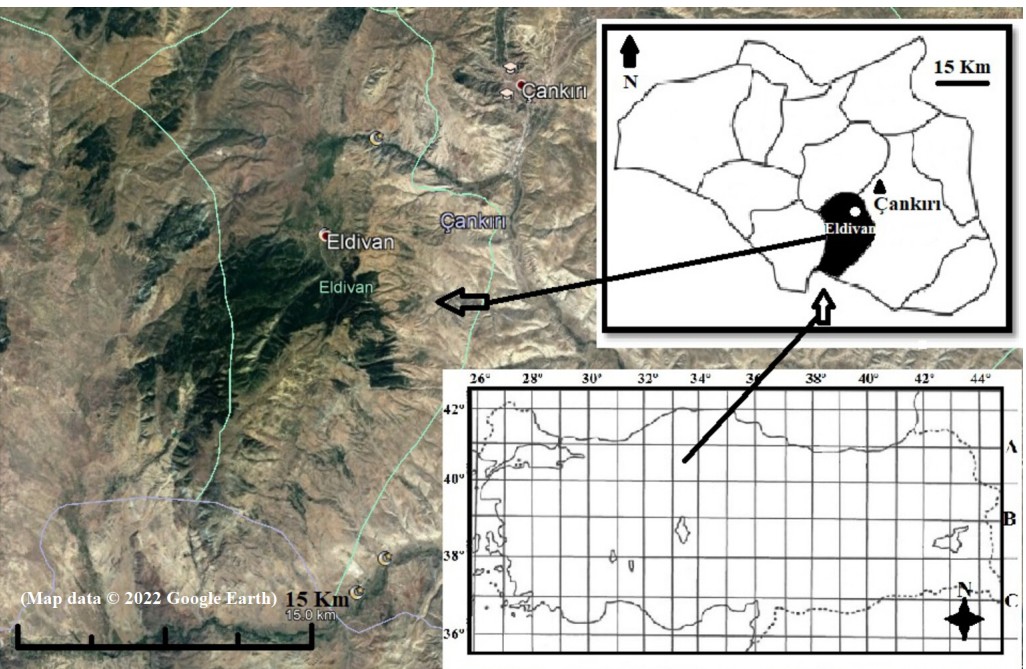

**Figure 1  Location of Eldivan Mountain, Turkey (Map data ©2022 Google Earth).**

but it has since emerged as the predominant species. Other notable tree species comprise *P. sylvestris* L., *Quercus pubescens* Willd., and *Q. robur* L.

The mountain falls inside the Eldivan district, physically positioned between 40°31′32″–40°26′16″N latitudes and 33°32′12″–33°24′02″E longitudes, with elevations varying from 1,000 to 1,810 m (*Ursavaş & Tuttu, 2020*). The Thornthwaite climate classification designates the region as having a ''semi-arid, mesothermal climate'' characterized by a mild winter water surplus, nearing a completely continental climate. Meteorological data from 1929 to 2021 reveal temperature extremes of 42.4 °C in July and −20.5 °C in March, alongside an average annual precipitation of 416.8 mm (*Ursavaş & Öztürk, 2016*; *OGM, 2021*).

## Sampling design

Initially, we identified 449 sampling points, but we excluded 37 due to insufficient forest coverage or proximity to anthropogenic features, like roads and streams. The exclusions were essential to concentrate the investigation on moss populations in forested areas, yielding ecologically pertinent data on biomass and dispersion. We excluded locations adjacent to streams, roads, or regions with inadequate forest cover, as these factors could introduce variability not pertinent to the basic research aims.

Existing bryophyte biomass studies, which highlight variability in sampling distances and plot sizes, guided a systematic approach to field sampling (*Peck & McCune, 1998*). For this study, a 300-meter sampling interval was adopted, as recommended by the ''Ecosystem-Based, Functional Forest Management Circular 299'' for densely forested

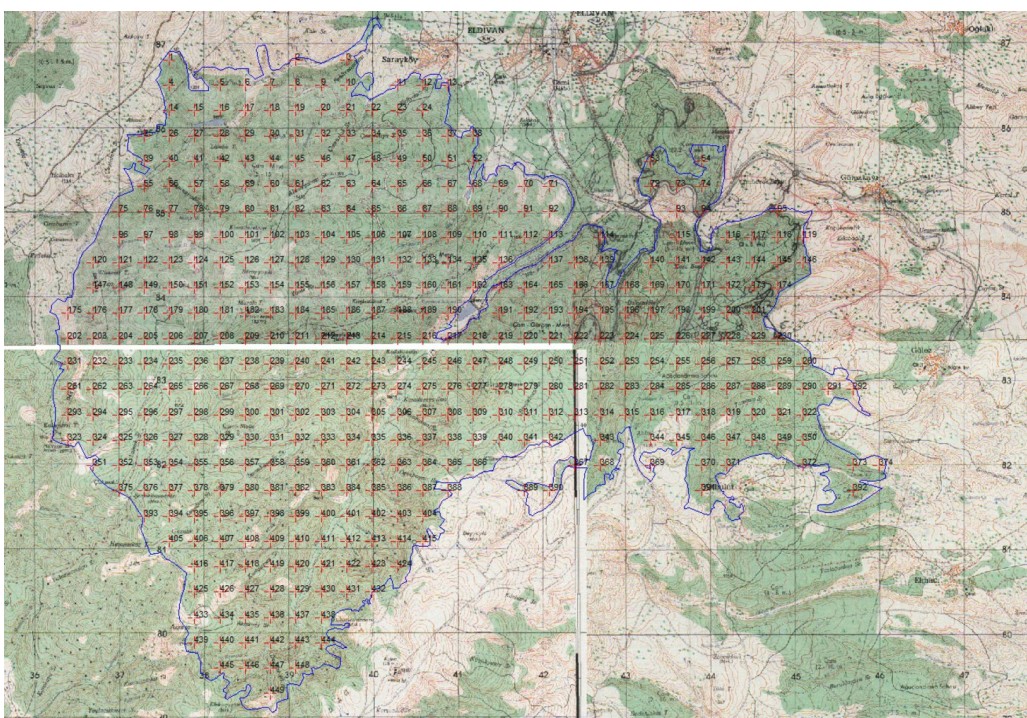

**Figure 2** Sampling points were taken at 300-meter intervals in the study area (Image created: ArcMap 10.3).

areas (*OGM, 2017*). We randomly selected initial sampling points across the extensive study area (4,200 hectares) and systematically positioned subsequent points using ArcMap 10.3 (Fig. 2). We established a total of 449 sampling sites across the research area.

To study ground mosses, two ×two m (four m$^2$) quadrats were used (*Piessens et al., 2008*; Fig. 3), and to study epiphytic mosses, 50 m$^2$ circular plots were used (*Caners, Macdonald & Belland, 2013*; Fig. 4). We relocated a sampling location up to 50 m to a representative site, if obstructions such as roads, streams, or bedrock made it unsuitable. We applied this adjustment in two cases: one steep incline and one road location.

We systematically arranged the quadrats, starting from the upper-left corner of the study area and proceeding sequentially from left to right. We used GPS devices to locate sampling points in the field, and ropes to outline two × two m quadrats. We photographed, measured, and collected each moss species within the quadrats in labeled plastic bags for laboratory analysis (Fig. 5).

In the laboratory, bryophyte samples were dried at 20–22 °C with 40–50% relative humidity for 1–2 days to prevent tissue deterioration (if the sample is very moist, the time may be extended). We performed morphological identification using an Olympus BX50 light microscope, focusing on features such as leaf structure, stem arrangement, and sporophyte traits. Identification relied on authoritative references (*Lawton, 1971*; *Nyholm, 1981*; *Greven, 2003*; *Smith, 2004*; *Heyn & Herrnstadt, 2004*; *Cortini, 2006*).

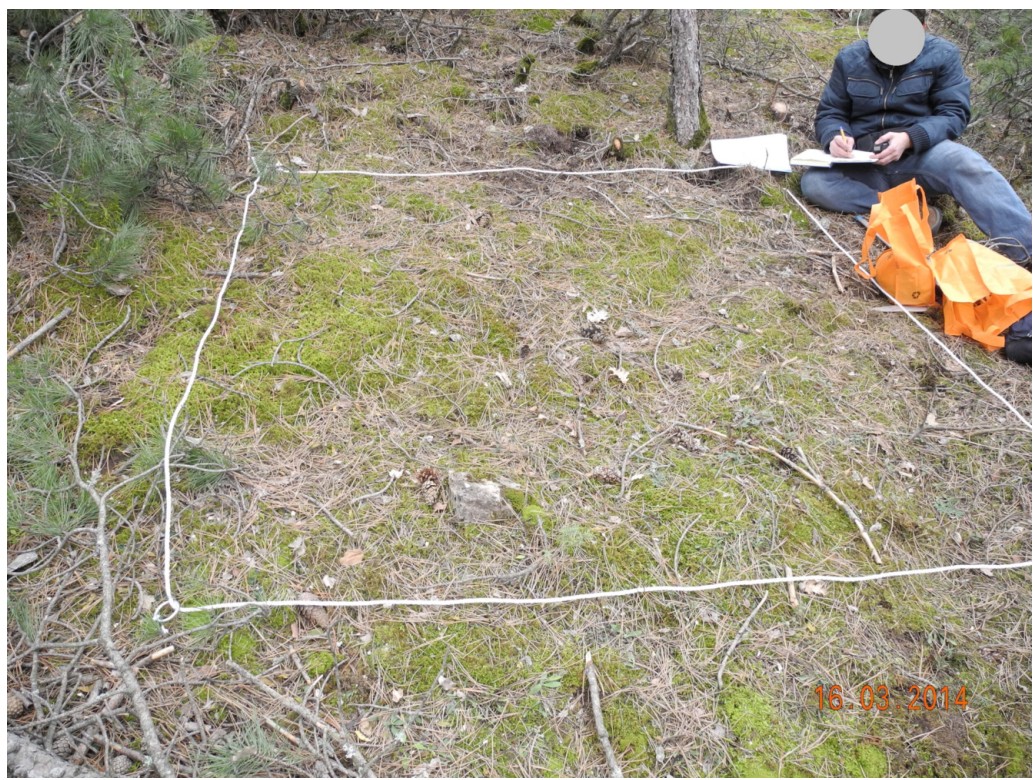

**Figure 3** Four square meter sampling area ($2 \times 2 = 4 \text{ m}^2$) (Photo credit: Recep SÖYLER).

We removed foreign materials such as soil and debris, and used a Kern ABT 320-4NM balance to measure moss samples with a precision of d: 0.0001 g.

The area covered by each moss species was calculated from the four m² quadrats and scaled to represent nine hectares. Dry weights were measured in grams for one × one, two × two, and four × four cm² segments and converted to kilograms to estimate biomass. (Pillow-shaped acrocarp mosses, due to their diminutive size, are segmented into one × one and two × two cm² pieces.). Pleurocarp mosses, which proliferate in a carpet-like manner, were segmented into four × four cm² pieces, and their dry weights were recorded. We proportioned biomass values to the total study area to estimate species-specific coverage and biomass.

## Data analysis

We conducted statistical analyses using SPSS version 20.0 (IBM Corp, Armonk, NY). Normality tests, including the Kolmogorov–Smirnov test, were applied to evaluate the distribution of the data. The results indicated that the dependent variable (dry weight) did not follow a normal distribution ($p < 0.05$). Consequently, non-parametric tests were used for further analyses.

The Kruskal–Wallis test, a non-parametric alternative to analysis of variance (ANOVA) examined relationships between moss biomass and environmental factors such as altitude, slope, and aspect. We identified significant differences between altitude categories and

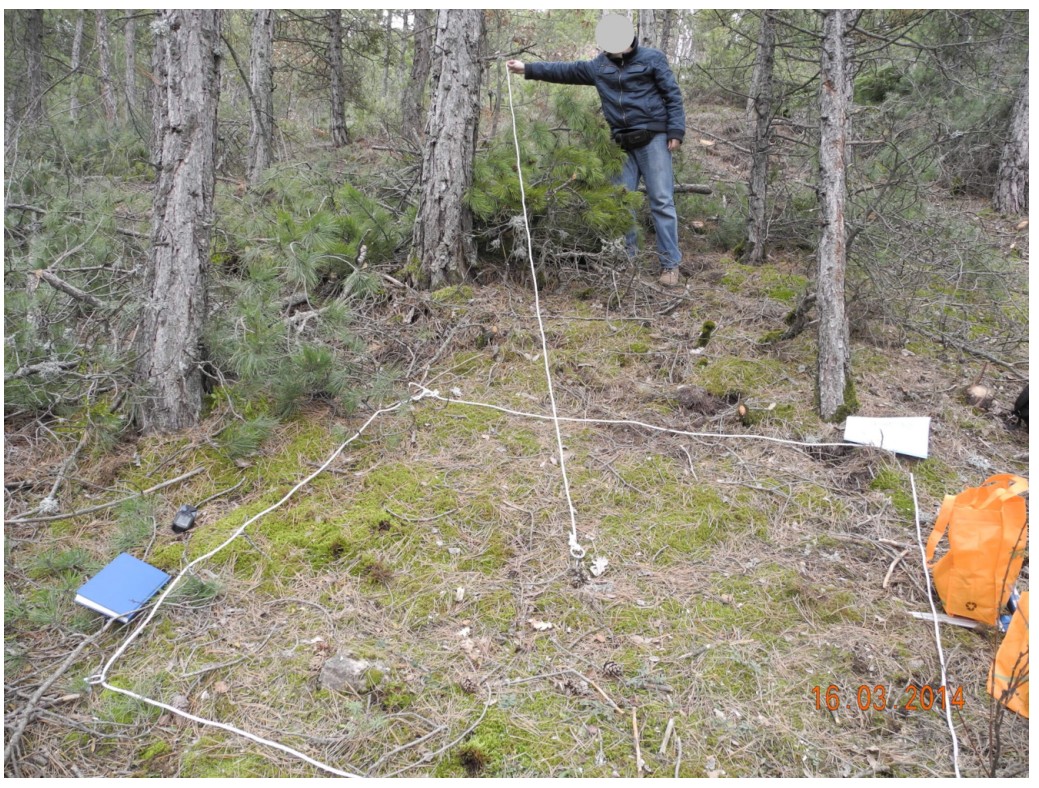

Figure 4  Sample area of 50 m² for epiphytic bryophytes on the tree (Photo credit: Recep SÖYLER).

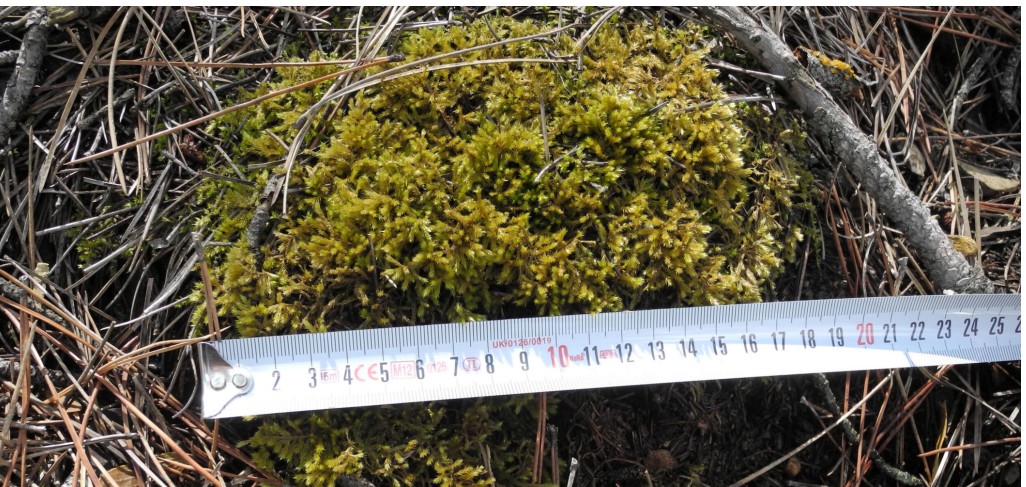

Figure 5  Each moss clump in a four m² area was measured separately (Photo credit: Serhat URSAVAŞ).

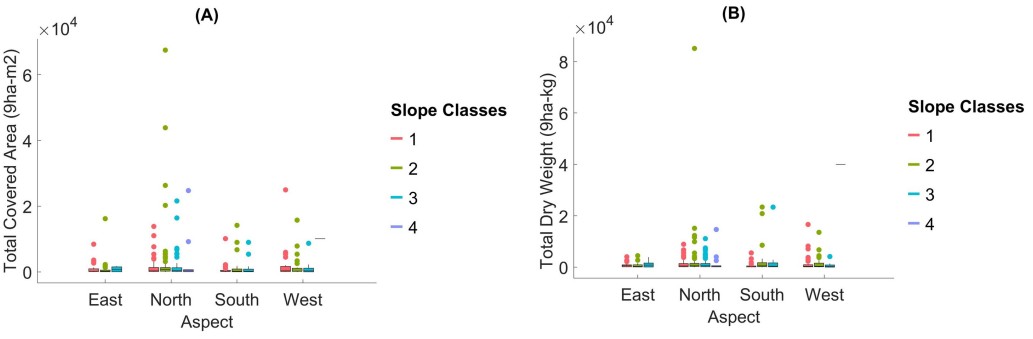

**Figure 6** The total covered area and the total dry weight of Mosses in terms of aspect and slope classes.

moss dry weight ($p = 0.042$), but found no significant differences for slope and aspect ($p > 0.05$).

We performed multiple linear regression analysis to assess the impact of continuous environmental variables. The regression model explained 30.1% of the variance in moss dry weight ($R^2 = 0.301$, $p < 0.001$). Area emerged as the most significant predictor ($\beta = 0.548$, $p < 0.001$), while slope had a minor, yet significant negative effect ($\beta = -0.107$, $p = 0.024$).

We classified topographical factors such as altitude, slope, and aspect into categorical groups for further analysis. We divided altitude into eight classes: Class 1 (1,000–1,099 m), Class 2 (1,100–1,199 m), Class 3 (1,200–1,299), Class 4 (1,300–1,399 m), Class 5 (1,400–1,499 m), Class 6 (1,500–1,599 m), Class 7 (1,600–1,699 m), and Class 8 (1,700–1,800). We divided the slope into four classes: Class 1 (flat, approximately 10%), Class 2 (moderate, 11–20%), Class 3 (high, 21–33%), and Class 4 (very steep, >33%). We categorized the aspect into four cardinal directions based on their azimuthal degrees: North (315°–44.9°), East (45°–134.9°), South (135°–224.9°), and West (225°–314.9°). These classifications facilitated comparisons of moss coverage and dry weight across varying environmental gradients. These classifications allowed for the calculation of total moss coverage (ha) and dry weight (kg) across categories, facilitating comparisons through Fig. 6.

## RESULTS

Fieldwork conducted at 412 sampling sites revealed the absence of bryophytes in 77 locations, particularly in degraded habitats and south-facing slopes, where limited sunlight and moisture hinder moss growth. Dense *Pinus nigra* subsp. *pallasiana* sapling communities further restrict light availability, which in turn impacts on moss photosynthesis. A total of 50 moss species were identified, including dominant species such as *Syntrichia ruralis*, and rare species like *Anoectangium aestivum, Encalypta streptocarpa, Pulvigera lyellii*, and *Syntrichia caninervis* var. *gypsophila* (located in a singular locality and covering a limited expanse). We analyzed data from 412 sampling points, revealing 28 genera from 13 families and 50 moss species. *Syntrichia ruralis* was the most widespread species, covering 67,016,378 m$^2$ with a dry weight of 734,119 kg. Other notable species included *Brachythecium erythrorrhizon*

and *Hypnum cupressiforme* var. *lacunosum* in arboreal habitats. Table 1 summarizes the distribution of all mosses, while Table 2 focuses only on epiphytic species.

## Distribution by aspect and elevation

The distribution of moss species varied significantly across different aspects.

North-facing slopes: Dominated by *Brachythecium erythrorrhizon* (18%) and *Dicranum scoparium* (15%). South-facing slopes: *Syntrichia ruralis* (42.11%) and *Brachythecium erythrorrhizon* (15%) prevailed. East-facing slopes: *S. ruralis* (20%) and *B. erythrorrhizon* (19%) were most common. West-facing slopes: *S. ruralis* (24%) and *Hypnum cupressiforme* (10%) were dominant.

Insufficient moisture in the semi-arid study area likely led to the identification of no liverwort specimens. *S. ruralis*, a shade-tolerant species, adapted well to the study area's microclimatic conditions.

## Effects of environmental variables

The statistical analysis results revealed the following:

Altitude: Using the Kruskal–Wallis test, altitude showed a significant effect on moss dry weight ($p = 0.042$), with a 95% confidence interval of (1.2, 3.5) kg for the mean ranks. The effect size ($\eta^2 = 0.04$) indicates a small, yet statistically significant impact. We observed the highest dry weight values in the 1,200–1,299 m and 1,400–1,499 m altitude intervals, suggesting that these ranges may offer optimal conditions for moss growth. We can attribute this effect to the moderate temperature and humidity levels in these zones. However, at very low (<1,200 m) and very high (>1,700 m) altitudes, dry weight values were reduced, likely due to less favorable environmental conditions.

Slope: The Kruskal–Wallis test did not detect significant differences in dry weight across slope classes ($p = 0.098$). This suggests that the slope may have a limited direct effect on moss biomass distribution, although steeper slopes are often associated with microhabitats that can retain moisture and reduce competition.

Aspect: Aspect showed no significant effect on moss dry weight ($p = 0.492$). This indicates that directional exposure may not strongly influence moss biomass in the study area, potentially due to the overriding impact of other factors, such as soil characteristics and canopy cover.

Regression analysis: Multiple linear regression analysis identified the area as the most significant predictor of moss dry weight ($\beta = 0.548$, $p < 0.001$), with a 95% confidence interval for the coefficient ranging from [0.40, 0.70]. Slope had a minor, yet significant negative effect ($\beta = -0.107$, $p = 0.024$, 95% CI [−0.20 to −0.01]). The model explained 30.1% of the variance in moss dry weight ($R^2 = 0.301$). Slope had a minor, yet significant negative impact on dry weight ($\beta = -0.107$, $p = 0.024$), suggesting that steeper slopes may slightly reduce biomass. The negative relationship between slope and moss dry weight ($\beta = -0.107$, $p = 0.024$) may be because of more erosion on steeper slopes, which makes it harder for mosses to grow and establish themselves by removing stable substrates. The model explained 30.1% of the variance in moss dry weight ($R^2 = 0.301$), indicating that other unmeasured factors may also contribute to biomass variability.

**Table 1** The distribution of moss species according to the areas they cover in the whole study area.

| Moss species | Red List status | Covered area (m²) | Dry weight (kg) |
|---|---|---|---|
| *Syntrichia ruralis* (Hedw.) F. Weber & D. Mohr | LC | 67,016,378 | 734,119 |
| *Abietinella abietina* (Hedw.) M. Fleisch. | LC | 10,245,200 | 19,860,166 |
| *Lewinskya rupestris* (Schleich. ex Schwägr.) F. Lara, Garilleti & Goffinet | LC | 2,141,650 | 5,215,816 |
| *Homalothecium sericeum* (Hedw.) Schimp. | LC | 1,880,639 | 1,199,937 |
| *Pseudoscleropodium purum* (Hedw.) M. Fleisch. | LC | 1,575,000 | 838,133 |
| *Brachythecium erythrorrhizon* Schimp. | LC | 1,438,227 | 632,625 |
| *Hypnum cupressiforme* var. *lacunosum* Brid. | LC | 1,287,230 | 705,483 |
| *Brachythecium albicans* (Hedw.) Schimp. | LC | 1,189,922 | 740,600 |
| *Hypnum cupressiforme* var. *cupressiforme* Hedw. | LC | 1,125,948 | 2,230,200 |
| *Homalothecium philippeanum* (Spruce) Schimp. | LC | 973,040 | 800,150 |
| *Brachythecium glareosum* (Bruch ex Spruce) Schimp. | LC | 945,000 | 478,566 |
| *Homalothecium lutescens* (Hedw.) H. Rob. | LC | 819,644 | 466,888 |
| *Syntrichia ruraliformis* (Besch.) Mans. | LC | 603,633 | 420,933 |
| *Tortula subulata* Hedw. | LC | 582,108 | 1,821,545 |
| *Rhytidiadelphus squarrosus* (Hedw.) Warnst. | LC | 525,000 | 761,133 |
| *Hylocomium splendens* (Hedw.) Schimp. | LC | 525,000 | 318,422 |
| *Dicranum scoparium* Hedw. | LC | 452,791 | 842,481 |
| *Brachytheciastrum velutinum* (Hedw.) Ignatov & Huttunen | LC | 411,211 | 458,681 |
| *Dicranum tauricum* Sapjegin | LC | 397,133 | 429,022 |
| *Encalypta rhaptocarpa* Schwägr. | LC | 378,000 | 830,200 |
| *Syntrichia virescens* (De Not.) Ochyra | LC | 364,700 | 923,533 |
| *Orthotrichum anomalum* Hedw. | LC | 225,400 | 140,350 |
| *Syntrichia norvegica* F. Weber | LC | 225,166 | 342,766 |
| *Amblystegium serpens* (Hedw.) Schimp. | LC | 223,844 | 191,800 |
| *Campylophyllopsis calcarea* (Crundw. & Nyholm) Ochyra | LC | 210,000 | 129,266 |
| *Lewinskya striata* (Hedw.) F. Lara, Garilleti & Goffinet | LC | 204,088 | 213,188 |
| *Tortella densa* (Lorentz & Molendo) Crundw. & Nyholm | LC | 178,500 | 127,866 |
| *Bryum* sp. Hedw. | | 157,266 | 321,766 |
| *Eurhynchium striatum* (Schreb. ex Hedw.) Schimp. | LC | 157,266 | 39,666 |
| *Syntrichia caninervis* var. *gypsophila* (J.J. Amann ex G. Roth) Ochyra | LC | 131,133 | 126,000 |
| *Tortella tortuosa* (Schrad. ex Hedw.) Limpr. | LC | 130,723 | 238,127 |
| *Grimmia funalis* (Schwägr.) Bruch & Schimp. | LC | 122,933 | 302,533 |
| *Ceratodon purpureus* (Hedw.) Brid. | LC | 112,233 | 443,333 |
| *Lewinskya affinis* (Brid.) F. Lara, Garilleti & Goffinet | LC | 109,128 | 110,851 |
| *Schistidium confertum* (Funck) Bruch & Schimp. | LC | 105,000 | 67,200 |
| *Ptychostomum imbricatulum* (Müll. Hal.) Holyoak & N. Pedersen | LC | 102,822 | 198,644 |
| *Lewinskya speciosa* (Nees) F. Lara, Garilleti & Goffinet | LC | 91,373 | 99,400 |
| *Heterocladiella dimorpha* (Brid.) Ignatov & Fedosov | LC | 84,000 | 84,000 |

*(continued on next page)*

**Table 1** (*continued*)

| Moss species | Red List status | Covered area (m²) | Dry weight (kg) |
|---|---|---|---|
| *Tortula marginata* (Bruch & Schimp.) Spruce | LC | 80,733 | 99,866 |
| *Grimmia trichophylla* Grev. | LC | 68,515 | 111,448 |
| *Syntrichia laevipila* Brid. | LC | 59,266 | 62,533 |
| *Tortula inermis* (Brid.) Mont. | LC | 53,822 | 56,933 |
| *Grimmia ovalis* (Hedw.) Lindb. | LC | 28,700 | 60,666 |
| *Gymnostomum calcareum* Nees & Hornsch. | LC | 26,133 | 102,666 |
| *Pulvigera lyellii* (Hook. & Taylor) Plášek, Sawicki & Ochyra | LC | 26,133 | 20,533 |
| *Grimmia pulvinata* (Hedw.) Sm. | LC | 16,893 | 28,140 |
| *Anoectangium aestivum* (Hedw.) Mitt. | LC | 16,800 | 133,933 |
| *Encalypta streptocarpa* Hedw. | LC | 9,333 | 62,533 |
| *Schistidium apocarpum* (Hedw.) Bruch & Schimp. | LC | 7,700 | 13,066 |
| *Tortula vahliana* (Schultz) Mont. | LC | 4,200 | 3,266 |
| TOTAL | | 97,216,557 | 44,640,972 |

**Table 2  Moss species detected only on the tree and their amounts in the whole study area.**

| Moss species | Covered area (m²) | Dry weight (kg) |
|---|---|---|
| *Hypnum cupressiforme* var. *lacunosum* Brid. | 3,937,266 | 1,448,533 |
| *Homalothecium sericeum* (Hedw.) Schimp. | 3,324,844 | 2,015,377 |
| *Brachythecium erythrorrhizon* Schimp. | 3,307,500 | 1,592,733 |
| *Syntrichia ruralis* (Hedw.) F. Weber & D. Mohr | 2,243,577 | 1,642,900 |
| *Hypnum cupressiforme* var. *cupressiforme* Hedw. | 2,083,783 | 1,089,666 |
| *Homalothecium philippeanum* (Spruce) Schimp. | 1,400,000 | 855,944 |
| *Orthotrichum anomalum* Hedw. | 840,000 | 505,866 |
| *Homalothecium lutescens* (Hedw.) H. Rob. | 826,700 | 408,566 |
| *Lewinskya striata* (Hedw.) F. Lara, Garilleti & Goffinet | 274,244 | 317,488 |
| *Brachytheciastrum velutinum* (Hedw.) Ignatov & Huttunen | 194,211 | 220,500 |
| *Dicranum scoparium* Hedw. | 120,691 | 328,883 |
| *Lewinskya affinis* (Brid.) F. Lara, Garilleti & Goffinet | 105,466 | 3,511,946 |
| *Lewinskya speciosa* (Nees) F. Lara, Garilleti & Goffinet | 73,344 | 78,477 |
| *Syntrichia laevipila* Brid. | 58,800 | 62,533 |
| *Tortula inermiş* (Brid.) Mont. | 28,233 | 14,233 |
| *Pulvigera lyellii* (Hook. & Taylor) Plášek, Sawicki & Ochyra | 26,133 | 20,533 |
| *Syntrichia virescens* (De Not.) Ochyra | 21,000 | 20,066 |
| *Grimmia pulvinata* (Hedw.) Sm. | 9,333 | 12,133 |
| *Amblystegium serpens* (Hedw.) Schimp. | 933 | 326 |
| TOTAL | 18,876,058 | 14,146,703 |

**Table 3  Regression results for moss dry weight, including 95% confidence intervals and effect sizes.**

| Predictor | Coefficient (B) | 95% CI | $\beta$ (Effect size) | p-value |
|---|---|---|---|---|
| Area | 0.548 | [0.40, 0.70] | 0.548 | <0.001 |
| Slope | −0.107 | [−0.20, −0.01] | −0.107 | 0.024 |

As shown in Table 3, altitude had a statistically significant effect on moss dry weight ($p = 0.042$), with a small effect size ($\eta^2 = 0.04$). This highlights the importance of altitudinal gradients in semi-arid ecosystems.

These results highlight the importance of altitude as a key factor influencing moss biomass, while slope and aspect appear to play more limited roles in the study area. Understanding these relationships can guide conservation strategies and sustainable harvesting practices.

## DISCUSSION

### Ecosystem contributions and conservation needs

Mosses contribute significantly to ecological processes, including soil stability and water retention (*Peck & McCune, 1998*). However, moss harvesting in Turkey occurs without sustainability considerations, risking over-extraction and biodiversity loss (*Ursavaş & Söyler, 2015*). Unregulated moss harvesting threatens ecosystem services, including soil stabilization and biodiversity (*Weber, Büdel & Belnap, 2014*). Sustainable harvesting practices must balance economic benefits with ecological preservation. Limiting harvesting to 1−1.5 tons per hectare, prioritizing the recovery of degraded areas, and protecting fragile habitats, like bedrock and epiphytic communities, are essential steps (*Atherton, Bosanquet & Lawley, 2010*). Numerous uncommon and fragile mosses are likely to inhabit rocks and crevices. These areas are home to species such as *Schistidium confertum*, *E. streptocarpa*, and *S. caninervis* var. *gypsophila*, which thrive in harsh conditions and require extra care due to their importance in maintaining biodiversity and halting erosion (*Peck & McCune, 1998*; *Muir, Norman & Sikes, 2006*).

A statistically significant effect size ($\eta^2 = 0.04$) and a 95% confidence interval of [1.2, 3.5] kg for mean ranks show that elevation does play a big role in how moss biomass is distributed. Although the effect size is small, it underscores the sensitivity of moss biomass to altitudinal gradients in semi-arid ecosystems. These findings highlight the necessity of considering altitude when developing conservation strategies and sustainable harvesting guidelines for mosses in Turkey.

Epiphytic mosses, which are more vulnerable in arid and semi-arid regions, depend on microhabitats with consistent moisture levels. The research area, with an annual precipitation of only 427 mm (*Ediş et al., 2022*), estimates the cumulative dry mass of mosses at 44,640,972 kg. This includes 14,146,703 kg from epiphytic species and 30,494,269 kg from terrestrial mosses. We calculated the sustainable harvestable biomass at 7,692,671 kg (1,831 kg/ha) after excluding rare species. Avoiding the collection of epiphytic mosses is crucial to maintaining ecosystem functionality and resilience in these water-limited environments (*Peck & Muir, 2008*).

## Biodiversity and habitat-specific conservation

The identification of 50 moss species highlights the biodiversity of Eldivan Mountain. Dominant species like *S. ruralis* play vital roles in moisture retention and nutrient cycling, particularly in semi-arid conditions (*Massatti, Doherty & Wood, 2018*; *Rosentreter, Bowker & Belnap, 2007*). Rare species such as *A. aestivum, E. streptocarpa, P. lyellii*, and *S. caninervis* var. *gypsophila* underline the need for habitat conservation, as these species are vulnerable to habitat loss and climate change (*Hodgetts & Lockhart, 2020*).

The relationships between moss species, slope exposure, and altitudinal gradients highlight the ecological preferences specific to different taxonomic groups. For example, moisture-dependent species like *Hypnum cupressiforme* thrive at higher elevations with cooler temperatures and higher humidity. Conversely, south-facing slopes with increased sunlight exposure limit moss growth due to higher evapotranspiration. These findings emphasize the importance of microclimatic variables in moss distribution and biomass, offering valuable insights for sustainable harvesting and habitat conservation.

Epiphytic mosses are found in restricted areas, primarily on roots and stumps, likely due to the low moisture availability in the semi-arid environment. The limited presence of species on rocky surfaces further emphasizes the importance of habitat-specific conservation strategies. Conservation measures must prioritize fragile habitats such as rocky outcrops, bedrock communities, and stream-adjacent zones, ensuring the survival of sensitive moss species and their ecological functions (*Ursavaş & Ediş, 2024*).

## Conservation and sustainability strategies

We propose the following recommendations to ensure sustainable moss harvesting while preserving biodiversity.

1. Selective harvesting: Focus on abundant species like *S. ruralis*, while avoiding rare species and mosses in sensitive microhabitats such as rocky outcrops and stream banks (*Peck & McCune, 1998*; *Muir, Norman & Sikes, 2006*).

2. Rotation harvesting: Divide the harvesting area into rotational zones, ensuring a minimum of five years for natural regeneration in each zone before re-harvesting (*Holling, 1973*; *Odum, 1985*).

3. Habitat protection: Limit harvesting in degraded areas and south-facing slopes with low moss density. Implement buffer zones of 15–20 m adjacent to streams and rocky outcrops, prohibiting moss harvesting to safeguard biodiversity and ecological functionality (*Ursavaş et al., 2021*; *Ursavaş & Ediş, 2024*).

4. Monitoring and data collection: Establish systematic monitoring programs to assess moss regrowth rates, species composition, and ecosystem health. Develop a database of moss species distribution and biomass metrics to guide adaptive management (*Peck & McCune, 1998*; *Ursavaş et al., 2021*).

5. Policy and community engagement: Put into effect national laws that demand inventory assessments prior to the issuance of harvesting permits. Engage local communities, forest managers, and policymakers to promote sustainable practices and establish certification systems for sustainably harvested moss products (*Demir, 2013*; *Studlar & Peck, 2007*).

6. Educational initiatives: Train harvesters in sustainable techniques, species identification, and ecosystem conservation practices to reduce the ecological impact of moss harvesting (*Peck & Moldenke, 2010*).

Incorporating these strategies into forestry management plans and regulatory frameworks can balance economic goals with ecological conservation. The preservation of uncommon and vulnerable species, especially in semi-arid regions, is essential to maintaining ecosystem resilience.

### Limitations and future directions

Environmental and methodological factors may influence biomass estimations:

Seasonal Variability: Changes in precipitation and temperature affect moss growth (*Peck & McCune, 1998*).

Anthropogenic Impacts: Land use changes and pedestrian activity may alter moss distribution.

Sampling Errors: Minor plot adjustments and measurement imperfections may introduce variability.

Future research should focus on:

1. Long-term monitoring to assess changes in species composition and resilience to environmental pressures.

2. Examining the effects of climate change on bryophytes distribution in semi-arid regions.

3. We can use genetic analyses to gain a deeper understanding of the adaptive strategies of both rare and dominant species. Future research should integrate genetic and physiological methodologies to enhance ecological insights and guide moss conservation measures. Genetic analysis can elucidate the diversity and structure of moss populations, facilitating the identification of at-risk groups and guiding focused conservation initiatives. Physiological investigations can evaluate the water retention, photosynthetic capabilities, and environmental tolerance of mosses, especially under climatic variability stress. Incorporating these data into ecological models might enhance sustainable harvesting protocols and forecast species distributions in future climate scenarios, ensuring conservation and sustainable utilization of moss resources.

4. We are expanding habitat suitability models by incorporating soil chemistry and canopy cover data.

These efforts will enhance conservation strategies and ensure the sustainability of bryophyte habitats and their ecosystem services.

## CONCLUSION

This study identified 50 moss species in Turkey's semi-arid forest ecosystems, analyzing their biomass and distribution. The results show how important dominant species like *Syntrichia ruralis* are for keeping the soil stable and moist, especially in semi-arid areas. We found that microclimatic factors like altitude and slope significantly influence moss distribution and biomass. These results underscore the importance of considering microclimatic variables for sustainable moss harvesting practices in semi-arid regions.

We discourage harvesting epiphytic mosses due to their sensitivity and dependency on limited water availability. According to field-based biomass analyses, we should limit sustainable harvesting of ground mosses to 1–1.5 tons per hectare to preserve ecosystem services and biodiversity. These findings provide a critical foundation for developing sustainable moss harvesting methods and conservation strategies aimed at balancing ecological preservation with economic benefits.

## ACKNOWLEDGEMENTS

My master's student Recep SÖYLER's thesis served as the basis for this study. We extend our sincere gratitude to Prof. Dr. İlker ERCANLI and Dr. Ferhat Bolat for their invaluable expertise and assistance with the Kruskal–Wallis and multiple linear regression analyses, which greatly contributed to the statistical rigor of this study. Additionally, we would like to thank Dr. Semih EDİŞ and Dr. Semih KUTER for their meticulous review and insightful feedback on the English language, which improved the clarity and readability of this manuscript.

### Funding

The authors received no funding for this work.

### Competing Interests

The authors declare there are no competing interests.

### Author Contributions

- Serhat Ursavaş conceived and designed the experiments, performed the experiments, analyzed the data, prepared figures and/or tables, authored or reviewed drafts of the article, and approved the final draft.
- Recep Söyler conceived and designed the experiments, performed the experiments, prepared figures and/or tables, and approved the final draft.

### Data Availability

Data is available at Zenodo:

Serhat URSAVAŞ. (2024). Evaluating Moss Diversity and Biomass for Sustainable Harvesting Methods in Semi-Arid Forests of Türkiye (Data set). In PeerJ. Zenodo. https://doi.org/10.5281/zenodo.14442075

### Supplemental Information

Supplemental information for this article can be found online at http://dx.doi.org/10.7717/peerj.19010#supplemental-information.

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
