# Peer review of "Evaluating moss diversity and biomass for sustainable harvesting methods in semi-arid forests of Turkey"

_PeerJ, doi:10.7717/peerj.19010_

## Round 0.1 · original submission · Major Revisions

Dear authors, I ask you to respond very carefully to each reviewer's comment. These responses should be accompanied by adding additional tables and graphs to the Results section of the article. The weak part of your manuscript is the statistical processing of the data. Perhaps you will find it possible to consult with specialists in statistical processing who will help improve this article. I believe that the large amount of field research you have done is worthy of publication, but now the manuscript needs very serious improvement.

Reviewer 1 ·

Basic reporting

The manuscript must be written in professional English, using clear and unambiguous language that is technically correct. It should fit within the broader field of knowledge, provide a clear interpretation of results, and be self-contained with relevant findings.

Experimental design

The research question should be clearly addressed, the knowledge gap should be well identified, and the methods should be described with sufficient detail.

Validity of the findings

The degree of advancement is not high, and while the novelty is interesting, replication studies are limited. The data should be robust and statistically sound, and the conclusion should be concise and specific regarding the key findings.

Additional comments

Comments for the authors
Title
The grammar and clarity of the suggested title are good, but here’s a slight adjustment for smoother readability:
"Estimating moss harvest quantity and biomass per hectare for sustainable harvesting in Türkiye"
Abstract
1. The sentence "This study, conducted on Eldivan Mountain over an area of 4,200 hectares, aimed to address this gap" could clarify why this topic is important by explaining the significance of addressing the gap in moss inventory and sustainable harvesting practices.
2. The sentence "Sampling points were established every 300 meters, measuring all mosses within 4 m² on the ground and 50 m² on trees" could be made more concise by rephrasing as "Sampling was conducted every 300 meters, measuring mosses in 4 m² ground plots and 50 m² tree plots."
3. The two sentences can be combined for clarity: "The most widespread ground species, Syntrichia ruralis, covered 64,772,801 m² with a dry weight of 623,268 kilograms, while the dominant tree species, Hypnum cupressiforme var. lacunosum, covered 3,937,266 m² and had a dry weight of 1,448,533 kilograms." Ensure that "dry weight" or "wet weight" is specified to avoid confusion.
Introduction
There are concerns about missing background knowledge. I suggest the authors address this in the introduction section. My recommendation for structuring the introduction is as follows: 1) Start with background information on moss ecosystems in Türkiye, followed by 2) the importance of moss harvesting, highlighting its ecological and economic significance, 3) challenges related to overharvesting and the need for sustainable practices, and conclude with 4) the purpose of the study: estimating moss harvest amounts and biomass to promote sustainable harvesting. Please consider these points when restructuring the introduction.
1. While the introduction touches on the importance of moss harvesting, it does not clearly state the knowledge gap specific to Türkiye. Emphasize what information is lacking about sustainable moss harvesting in Türkiye and why this is important compared to regions like the U.S. and Europe.
2. Since the global demand for Non-Timber Forest Products (NTFPs) is mentioned, provide more context about how Türkiye fits into this market. How significant is moss harvesting in Türkiye compared to other countries? Are there unique challenges related to its semi-arid climate?
3. To streamline the literature review, improve the flow of the introduction by starting with a broader overview of moss harvesting globally, and then narrowing the focus to the U.S. and Türkiye. Group similar findings to avoid repetition.
4. The lack of inventory studies on moss in Türkiye is mentioned, but further elaboration on the ecological, economic, and environmental implications of this gap would be helpful.
5. While the objectives are mentioned at the end, providing a brief explanation of how the research will meet each objective would improve clarity. For example, explain how determining biomass or identifying rare species will contribute to sustainability efforts. Additionally, the objectives outlined by the authors are unclear regarding the "amount of bryophyte biomass, in line 60" as the authors measure the areas of bryophyte cover and the dry weight of bryophytes. Do the authors interpret bryophyte biomass using two different terms, i.e., the area of bryophyte cover and dry biomass? Please clarify.
6. Begin the introduction with a general discussion about NTFPs and moss harvesting, and then narrow down to the specific context of Türkiye. This approach will create a clear progression for the reader and better frame the research within a broader context.
7. The introduction mentions the ecological role of bryophytes, but the authors could further explain their importance in ecosystem services, such as moisture retention and soil stabilization, to reinforce the environmental relevance of the study.
I suggest that the authors move the 'Research Area' section [lines 67-82] to combine it with the first paragraph of the Materials and Methods section [lines 87-95], along with its associated Figure 1. This combined section can stand alone under the title "Study Area and Sampling Design." Additionally, when writing a scientific name for the first time, the genus name should be spelled out in full (e.g., Pinus nigra) [line 69]. In subsequent mentions, it can be abbreviated (e.g., P. nigra in line 88). The same applies to other species names, such as Quercus pubescens Willd. [line 70] and Q. robur [line 70].
Materials and Methods
Overall Comment: Review this section for clarity and flow. Avoid overly complex sentences and jargon unless necessary. Consider breaking long sentences into shorter ones for better readability. Other constructive comments are as follows:
1. The second-to-last paragraph of the Materials and Methods section [lines 100-134] could be separated to stand alone as its own section titled "Sampling Methodology and Data Collection."
2. Use clear subheadings within the Materials and Methods section to help organize the content more effectively. For instance: Study Area, Sampling Design, Data Collection, and Data Analysis.
3. While you mention Çankırı Eldivan Mountain [lines 88-89], consider adding geographic coordinates or a brief description of the area’s ecological significance.
4. Could the authors elaborate on the sampling rationale? For example, explain why specific sampling intervals (e.g., 300 meters) [line 90] were chosen and their relevance to moss population studies. Additionally, provide more specifics on how the quadrat sampling was conducted. For instance, how were the quadrats placed? Were they positioned randomly or systematically?
5. Could the authors specify the instruments and techniques used? For example, provide details about the types of instruments (e.g., tree meters) selected for measurements and the rationale behind their selection. Additionally, clarify how length, width, and area calculations were made; for instance, explain how cm² was converted to m² for reporting. Regarding "Figure 4. Measurement of the width and length of the moss samples" in line 113, do the authors mean the measurement of the moss samples themselves, or the area of moss cover? Please clarify.
6. How did the authors handle the unsuitable locations mentioned in line 115? Please clarify the adjustment protocol. For example, specify how frequently unsuitable sampling locations occurred and the criteria used to determine whether a location was suitable.
7. Could the authors provide more details about the drying process for bryophytes and the methods used for species identification in line 126? Include information about the drying conditions in line 132, such as temperature and humidity, to ensure reproducibility by other researchers. Additionally, specify the methods used for identifying moss species (e.g., morphological identification or genetic methods).
8. Please authors include a brief statement explaining what skewness and kurtosis values indicate about data distribution [lines 138-139], as this will help in understanding the subsequent analysis.
Other revisions to consider include the following:
• It might be helpful to incorporate more robust normality tests, such as the Shapiro-Wilk or Kolmogorov-Smirnov tests, to further strengthen the analysis.
• Since the data exhibited a normal distribution, using parametric tests like ANOVA seems reasonable. However, ensure that other assumptions of ANOVA, such as homogeneity of variances (which can be tested using Levene's test), are also verified.
• ANOVA is appropriate for examining differences between groups (e.g., moss species and environmental factors such as altitude, aspect, area covered, and air-dry weights). However, if multiple factors are being considered (e.g., altitude and aspect), a multivariate analysis (e.g., MANOVA) or multiple regression analysis may be more suitable, especially if the goal is to explore the combined effects of these variables on moss species distribution.
• If the authors' focus is more on understanding relationships between variables rather than detecting differences, techniques such as correlation analysis or regression models might offer deeper insights. For example, linear regression or generalized linear models (GLM) could provide more detailed information regarding the strength and direction of the associations between moss species and environmental variables.
Results
Overall Flow:
To improve the flow, consider reorganizing the section as follows:
• Start with a brief overview of the sampling process and the reasons for excluding certain points.
• Provide a detailed interpretation of the key findings from the tables.
• Present statistical results (ANOVA and post-hoc analysis), ensuring all assumptions and findings are clearly explained.
Specific Points of Clarification
1. Introduce the reason for the reduction of 37 points upfront in line 147 to ensure clarity regarding the sampling process. The authors might want to elaborate on the ecological and methodological significance of excluding certain points, for example: "The points were excluded to maintain consistency in sampling forest-covered areas for studying moss populations."
2. The authors refer to tables but do not interpret their contents in detail. It's important to summarize key findings from Table 1 and Table 2 rather than only mentioning their existence. For instance: "Table 1 shows the distribution of taxa according to the areas covered, highlighting that species X dominated these areas, particularly in region Y. Table 2 focuses on epiphytic moss species, demonstrating that species Z were more abundant in tree-covered areas."
3. Consider rephrasing lines 161-166 to improve flow. Additionally, these rephrased sentences should be moved to the discussion section: "The absence of bryophytes was primarily observed in degraded stands and south-facing aspects, likely due to restricted sunlight penetration in these areas. Dense Pinus nigra subsp. pallasiana sapling communities may have limited light availability, negatively affecting moss growth and photosynthesis."
4. What do the authors mean by “the sigma” in line 168? Once again, I believe the appropriate test for the objectives mentioned by the authors is a correlation test, such as assessing correlations between factors like:
o The number of taxa & altitude
o The number of taxa & slope
o The number of taxa & aspect
o Area covered & altitude
o Area covered & slope
o Area covered & aspect
o Sample (dry?) weight & altitude
o Sample (dry?) weight & slope
o Sample (dry?) weight & aspect
5. I did not see Table 3 correspond to any explanation in the text; it appears to be merely a table of descriptive statistics but does not address what the authors mentioned in line 169. Additionally, the data in Table 3 are not presented in a suitable format for the article (e.g., the biggest and the smallest values). What do the authors mean by the biggest and smallest? What does the number of samples in Table 3 signify? Does it refer to the number of sampling points? I tried to relate this to the explanation described in Materials and Methods, where it states that the number of sampling points is just 412 (n=412), but the number of samples in Table 3 exceeds 412.
6. The authors mention that elevation, aspect, slope, and station factors showed statistical significance in the ANOVA test in Table 4. However, more clarity is needed on how these factors affected moss species richness (number of taxa), area covered (ha), and sample (dry?) weight. From the table, it appears that the ANOVA tests are significant for the shaded cells.
7. The authors presented Figure 6, although there were no differences between 1) the total covered areas in the four different aspects (East, West, North, and South) and 2) the total air-dry weight in the four different aspects (East, West, North, and South), as shown in Table 4. Why is this the case? What about the ANOVA test for altitude and slope with respect to the number of taxa, area covered, and sample weight? How many categories are there for altitude (e.g., low, medium, high) and slope (e.g., flat, steep, very steep)? All of these factors should be clarified and mentioned in the Materials and Methods section.
8. When mentioning significant relationships between altitude, aspect, and moss taxa (Table 4), further detail the biological significance of these findings. Explain how the observed relationships might relate to the ecological preferences of the moss species (e.g., are some species more common at higher altitudes due to humidity levels?).
Discussion
General Structure
• Begin the discussion with a brief recap of the main findings to remind readers of the study's key results. For instance, mention the dominant species and any significant patterns in distribution at the outset.
• Consider dividing the discussion into clear subsections (e.g., Species Diversity, Ecological Significance, Conservation Implications) to enhance readability.
Content Improvements
1. Provide more context on the global significance of your findings. Explain how your results relate to existing literature on moss biodiversity and ecology.
2. In the Results section [lines 162-163], the authors mention the absence of bryophytes in some regions (degraded stands and south-facing areas). It would be more insightful to further discuss possible ecological reasons for their absence in the Discussion section. For example, link the absence to specific environmental stress factors or land-use patterns, and explain how these conditions influence moss distribution.
3. Discuss the implications of your findings in light of the limited research on moss in Turkey and the broader ecological context.
4. Expand on the ecological role of dominant species like Syntrichia ruralis. Discuss how this species contributes to ecosystem functions, such as soil stability and moisture retention.
5. Address the potential conservation implications of over-harvesting moss. Offer specific recommendations for sustainable harvesting practices based on your findings.
6. Clearly explain any statistical results that support your claims. If there are significant differences or trends in the data, describe these in detail to underscore their importance.
7. For instance, when discussing the ANOVA results, specify which factors were significant and how they relate to species richness or biomass investigated by the researchers.
8. Be consistent with the terminology used. For example, clarify whether “dry weights” refer to air-dry weights [lines 228-230] and ensure this definition is consistently applied throughout.
9. Some sentences are repetitive or overly complex. For example, instead of stating "the least abundant moss species in the study area" in line 219, consider stating directly: "The least abundant species include Tortula vahliana, Schistidium apocarpum, and Encalypta streptocarpa."
10. Break down long sentences into shorter, more digestible ones. For example, instead of saying, "These species typically colonize rocky surfaces or crevices within the forest" in lines 221-222, consider: "These species generally colonize rocky surfaces. They thrive in crevices within the forest."
11. If relevant, compare your findings to those from similar studies in different geographic areas. This comparison can highlight the uniqueness or commonality of your study area.
12. Conclude the discussion with suggestions for future research, which could include the need for long-term monitoring of moss populations or studies exploring the effects of climate change on these bryophyte species.
Conclusion
1. The section can be made more coherent by following a logical structure that clearly presents the context, findings, and recommendations.
2. While the analysis includes valuable data, it would be helpful to provide a brief overview of the significance of moss ecosystems in Turkey and the potential impacts of harvesting practices on these ecosystems.
3. Ensure that terms and measurements are used consistently throughout the section. For example, clarify what “air-dry weight” means in this context [lines 260-261].
4. The numerical data should be presented accurately; please verify their correctness [lines 259-260].
5. The recommendation against harvesting epiphytic species should be emphasized and briefly explained. You might phrase it as, "Given the low precipitation levels in Turkey, it is crucial to avoid harvesting epiphytic species, as they are more susceptible to environmental stressors."
6. Expand on the implications of harvesting from bedrock. Explain why it is essential to protect rare and sensitive species that may inhabit these areas, potentially including examples or more detailed reasoning.
7. Conclude with a strong statement that summarizes the importance of sustainable practices for moss harvesting and encourages further research or monitoring efforts.

Reviewer 2 ·

Basic reporting

The rationale for collecting the dataset is briefly mentioned but could be more clearly articulated. It should emphasize the utility of the dataset beyond this study to the broader research community. For example, how can the data be applied in other geographical areas or environmental contexts?

Suggestion: Expand the introduction to include potential uses for the data in conservation, policy-making, or ecosystem management.

While data was collected and reported, the paper does not provide a clear mention of depositing the dataset in a public repository. For PEERJ standards, data must be uploaded or linked to a publicly accessible repository with a DOI.

Suggestion: Upload the dataset to a repository like Dryad or Zenodo, ensure it’s available under an open license, and include a DOI in the paper for future reuse by other researchers.

Experimental design

If the dataset is expected to be extended over time, this paper does not discuss plans for ongoing data collection or curation, which could be critical for long-term moss conservation efforts.

Suggestion: Include a section discussing whether this dataset will be updated periodically and provide details on future curation efforts (e.g., expansion to other regions, improvements in techniques).

The exclusion criteria for sampling points (e.g., locations near roads or degraded areas) are mentioned but lack clear explanations on how these exclusions might impact the generalizability of the findings.

Suggestion: Provide more justification for exclusion criteria and discuss potential biases these choices introduce to the dataset.

Validity of the findings

The paper does not adequately discuss the limitations of the dataset. For instance, how do environmental variables like climate fluctuations or human activity potentially skew the data?

Suggestion: Add a dedicated limitations section that highlights factors that could affect the accuracy of biomass estimation (e.g., seasonality, sampling errors).

While descriptive statistics are included, the paper borders on analyzing the dataset in a way that suggests testing hypotheses (e.g., correlation between slope and moss distribution). According to PEERJ’s standards, if you are testing hypotheses, it should be submitted as a research article rather than a data report.

Suggestion: Ensure the paper maintains its focus on reporting the data rather than extending into hypothesis testing, or consider restructuring the submission as a research article.

Additional comments

Overall, the paper presents a well-structured dataset with practical implications for sustainable moss harvesting.

Annotated reviews are not available for download in order to protect the identity of reviewers who chose to remain anonymous.

Reviewer 3 ·

Basic reporting

Few studies on moss harvesting have been reported. Authors did a tremendous work, however, the experiment design and data analysis are too simple and the writing is poor. The results only provided moss harvest amount and biomass per hectare and the scientific questions are not weil addressed.

Experimental design

Biomass recovery is essential to evaluating sustainable harvesting; however, authors did not design any relevant experiment.

Validity of the findings

1. Authors should make clear the general importance of their findings.
2. The section of methods should be reorganized. It is a tremendous field work. However, how many samples collected are unknown.
3. Authors analyzed the relationships of species and variables in Lines 140-143. What’s the significance of this analysis?
4. Figs. 2-5 are unnecessary.
5. Table 3 and Table 4 seem to be direct from the original results of statistical software.

Additional comments

1. It is not clear about “sustainable harvesting”. How to define or evaluate the “sustainable” harvesting? Is there any related ecological theory?
2. Although there are few studies on moss harvesting, it is necessary to review the related research and is not just about the harvesting weight of mosses.
3. I strongly suggest that authors indicate the roles of mosses or bryophytes in ecological functioning of ecosystems and commercial moss species, which would help readers understanding the importance of sustainable moss harvesting.
4. Some paragraph belong to the section of “Results”, such as Lines 238-250. Even moving to the section of “Results”, it should be rewritten.

---

## Round 0.2 · Major Revisions

Dear authors, I ask you to carefully correct the manuscript in accordance with the reviewers' comments. Ignoring the comments or incomplete response to them may result in the article being rejected.

Reviewer 1 ·

Basic reporting

The manuscript has been significantly improved, particularly in the Introduction section, which now provides valuable insights into moss biomass estimation and sustainable harvesting practices. However, several clarifications are still needed regarding the statistical methodology, consistency in data presentation, and overall manuscript formatting to ensure scientific clarity. Addressing the concerns outlined below will enhance the manuscript's impact and adherence to scientific integrity.

Experimental design

Concerns and Recommendations
Statistical Tests
1. Appropriateness of ANOVA
• ANOVA is suitable for categorical independent variables, but the manuscript does not clearly describe how continuous variables such as altitude and slope were categorized (e.g., into ranges or intervals). If these variables were not categorized, regression analysis would be a more appropriate approach.
• Ensure that ANOVA assumptions (normality, homogeneity of variance, and independence of data points) were tested and reported. While normality is mentioned, the homogeneity of variance (e.g., Levene's test) has not been explicitly addressed.
2. Regression Analysis
• Regression analysis is more appropriate for continuous predictors, as it provides deeper insights into the relationships between variables such as altitude, slope, and moss biomass.
• Include details about the robustness of the regression models, such as R² values and multicollinearity checks.
3. Data Presentation
• Statistical results should include confidence intervals and effect sizes to enhance the interpretation of the significance and impact of the variables.

Validity of the findings

Figures and Tables
• Figures:
o Ensure that the captions for Figures 4 and 5 accurately describe their content.
• Tables:
o Ensure consistency in the presentation of units (e.g., m², kg) and clearly separate numerical values from their labels for easier interpretation.

Additional comments

1. Manuscript Structure
• Results:
o Move the “Ecological Implications” section to follow the Discussion section.
• Discussion:
o Statistical test results currently included in the Discussion section should be relocated to the Results section.
• Conclusion:
o Make the Conclusion more concise, focusing on the key findings and their implications for sustainable moss harvesting. Currently, it reads more like a continuation of the Discussion section.

2. Technical and Language Issues
• Revise the manuscript to correct minor grammatical issues and improve readability. For example:
o "Sampling points taken at 300 m interval" → "Sampling points were taken at 300-meter intervals."
o Replace "altitudes" with "elevations" for consistency in geographic terminology.
• Seek professional language editing to enhance clarity for an international audience.
• Correct typographical errors, such as "Moses" instead of "Mosses."

3. Additional Suggestions
• Sustainability Recommendations:
o Provide detailed suggestions for implementing sustainable harvesting practices, including policy recommendations or guidelines for rotational harvesting.
• Future Research Directions:
o Elaborate on how genetic or physiological studies could complement the findings and inform future moss conservation strategies.

·

Basic reporting

The article is based on considerable factual material. The content of the article is of interest to a wide range of readers, as it deals with issues important to plant biology.
However, the article has some peculiarities that should be corrected before publication.

Experimental design

Satisfaisant

Validity of the findings

Satisfaisant

Additional comments

1. The title of the article refers to the practical aspect of the results obtained in the work, but not to the work as a whole. The dependence of diversity and biomass of moss communities is the main content of the article. Practical aspects are very important, but the title of the article should indicate what the essence of the study is.
2. Style of presentation: the article consists of parts that sometimes do not correlate well with each other in terms of text style and sequence of presentation. This is to some extent the case with the literature review. In the Results and Discussion sections, the text has the character of ‘notes in the margins’ and cannot be the final version of the manuscript.
3. The search for the dependence of environmental indicators on environmental factors (elevation, slope, aspect) is an example of a direct gradient analysis. Monotonic dependence is not the only or the most common variant of such dependence. Therefore, the mathematical tools of the study require some improvement.

·

Basic reporting

Abstract:
Lines 24-26. At the first mention of the Latin name of a living organism in the abstract and the text of the manuscript (lines 181, 183, etc.), the author's surname and the year of description of the species must be indicated without abbreviations.

Introduction:
Lines 48-51. Statements must be supported by a citation.

Results:
Lines 175-180. The sentences belong to the Materials and Methods section. Move them.
Line 181. What do you mean by “…50 moss taxa”? Genera and families of mosses are also taxa. Are they species? In the International Code of Nomenclature for algae, fungi, and plants (2018), the term “taxon” refers to a taxonomic group of any rank.
In table titles and throughout the manuscript, replace the term “taxa” with “species.”
Lines 217, 221. Citations are not allowed in the Results section. It is generally recommended to place limitations of studies in the Discussion section.
Lines 224-225. Rephrase the sentence: “The connections between moss taxa, aspect, and altitude underscore species-specific ecological preferences.” Perhaps the authors meant taxonomic affiliation, slope exposure and altitudinal gradient?

Experimental design

Materials and methods:
Line 127. Separate literary references from the sentence.
Line 128. I recommend removing the phrase "... a precision balance with 0.000 g accuracy". The concept of scale accuracy is very relative. The accuracy of electronic devices is affected by many factors: temperature, humidity, surface type, human factor, etc. Therefore, it is better to indicate the model name and discreteness (d) (measurement step) of the laboratory scale.

Validity of the findings

Discussion:
Lines 233, 257. Remove subparagraphs. Research results should be in the "Results" section.
Lines 234-240, 258-266. Move to the "Results" section.

Conclusion:
Lines 305-321. Move the text to the "Discussion" section. Citing literary sources in the "Conclusion" section is not allowed.

Additional comments

No comments.

---

## Round 0.3 · Major Revisions

Dear authors, I agree with the reviewer. Why are there references to literature in the Conclusion section? I also draw your attention to the spelling of geographical names, which should be in English, not Turkish (not only in the title of the article, but also in the places of your work, and in the text of the article). Lines 173-177, 274-371: please study how this information is given in articles already published in our journal. This information cannot be published in this form. There are many technical errors in the text. Please involve your more knowledgeable colleagues in editing the article. I hope that you will carefully correct these comments and this will not give me the opportunity to refuse you publication.

·

Basic reporting

All recommendations of the previous review have been addressed. I recommend the article for publication

Experimental design

All recommendations of the previous review have been addressed. I recommend the article for publication

Validity of the findings

All recommendations of the previous review have been addressed. I recommend the article for publication

Additional comments

All recommendations of the previous review have been addressed. I recommend the article for publication

·

Basic reporting

The authors took into account most of my comments and made changes to the manuscript. However, they ignored the last comment:
Referencing literary sources in the "Conclusion" section is not allowed.

Experimental design

No comments.

Validity of the findings

No comments.

Additional comments

No comments.

---

## Round 0.4 · accepted · Accept

Dear authors, I am pleased to inform you that the reviewers have approved the publication of your article and it has been accepted for publication. I wish you further success in your scientific work and further good publications!

·

Basic reporting

The authors took into account all my comments and made changes to the text of the manuscript. The article can be recommended for publication.

Experimental design

The authors took into account all my comments and made changes to the text of the manuscript. The article can be recommended for publication.

Validity of the findings

The authors took into account all my comments and made changes to the text of the manuscript. The article can be recommended for publication.

Additional comments

No comments.